# Is SARS-CoV-2 a Risk Factor of Bipolar Disorder?—A Narrative Review

**DOI:** 10.3390/jcm11206060

**Published:** 2022-10-14

**Authors:** Piotr Lorkiewicz, Napoleon Waszkiewicz

**Affiliations:** Department of Psychiatry, Medical University of Bialystok, Wołodyjowskiego 2, 15-272 Białystok, Poland

**Keywords:** COVID-19, bipolar, mania, depression, kynurenine, cytokines, oxidative, corticosteroids

## Abstract

For 2.5 years we have been facing the coronavirus disease (COVID-19) and its health, social and economic effects. One of its known consequences is the development of neuropsychiatric diseases such as anxiety and depression. However, reports of manic episodes related to COVID-19 have emerged. Mania is an integral part of the debilitating illness—bipolar disorder (BD). Due to its devastating effects, it is therefore important to establish whether SARS-CoV-2 infection is a causative agent of this severe mental disorder. In this narrative review, we discuss the similarities between the disorders caused by SARS-CoV-2 and those found in patients with BD, and we also try to answer the question of whether SARS-CoV-2 infection may be a risk factor for the development of this affective disorder. Our observation shows that disorders in COVID-19 showing the greatest similarity to those in BD are cytokine disorders, tryptophan metabolism, sleep disorders and structural changes in the central nervous system (CNS). These changes, especially intensified in severe infections, may be a trigger for the development of BD in particularly vulnerable people, e.g., with family history, or cause an acute episode in patients with a pre-existing BD.

## 1. Introduction

The coronavirus disease (COVID-19) caused by the SARS-CoV-2 virus has led to a worldwide pandemic, and over the past 2.5 years, controlling it has been one of the biggest challenges which modern medicine has faced. To this date, according to WHO data, 528 million people have been infected, and almost 6.3 million people have died as a result [1]. Although we currently have effective methods of detecting the virus and a vaccine that significantly reduces the severity of its symptoms and lowers mortality [2], the health, social and economic consequences of the pandemic remain a major problem. One of its health consequences are the already frequently described psychiatric disorders resulting directly from the influence of the SARS-CoV-2 virus on the body as well as from the social aspects of the pandemic such as isolation, fear of disease, financial problems and the overall stress caused by the pandemic [3,4,5]. For the clarity of our considerations, in the following work we will focus primarily on the direct impact of the infection.

Until now, the most frequently described psychiatric disorders in connection with COVID-19 are major depressive disorder (MDD), generalized anxiety disorders (GAD), post-traumatic stress disorder (PTSD), obsessive compulsive disorder (OCD), sleep disorders, cognitive and memory disorders [4,5]. MDD is the major neuropsychiatric outcome of COVID-19, with an incidence ranging from 4% to 12% [4,6], and shares many biochemical and immunological disturbances with SARS-CoV-2 infection [7] Since 2.5 years is still a short time, it would be reasonable to assume that some of the long-term effects of infection with SARS-CoV-2 virus are still a mystery to us and they will manifest themselves in the future [8]. Therefore, it is crucial to understand the possible mechanisms that influence their development, and which can be the starting point for choosing the appropriate treatment and prevention strategies. The analysis of changes taking place in the body under the influence of SARS-CoV-2 may also allow us to predict what effects of the disease can be expected in the future, as well as to better understand the development of psychiatric disorders in general.

Bipolar disorder (BD) is a type of affective disorder characterized by recurring episodes of depression and mania or hypomania with possible psychotic symptoms. It most often begins in young people and affects men and women equally [9,10]. It has a significant impact on the quality of life, its mental and physical zone. Depending on the research, the lifetime prevalence ranges from 0.1 to 2.4% [11]. The risk of suicide in patients with BD is approximately 4× higher than in patients with MDD and 10–30× higher than in the general population. Up to 60% of BD patients report suicidal thoughts, 50% attempt suicide, and about 20% commit it [12,13,14]. Additionally, the risk of death due to other diseases, such as cardiovascular, metabolic or respiratory diseases, is greater in patients with BD than in the general population [15]. The key elements of BD are manic or hypomanic episodes and they are characterized by increased mood, hyperactivity, impulsiveness, grandiosity, increased self-esteem, decreased anxiety, decreased need for sleep, increased sex drive and in some cases psychosis [16].

The etiopathogenesis of this disease is still largely unknown, although there are many hypotheses about it that take into account factors such as inflammation and its markers, changes in genetics and epigenetics, metabolism, hormones, neurotransmission and immunology as well as oxidative stress and external stressors [17,18,19,20]. This disorder is neuroprogressive, i.e., in its course, there is a gradual clinical and functional deterioration of the patients’ condition [17,19]. Changes in neuroplasticity, deficiencies in neurotrophic factors, oxidative stress, inflammation and circadian rhythm disturbances are associated with exacerbation of affective episodes, treatment resistance, and cognitive and functional dysfunctions [21,22,23]. Despite all the data collected, a consistent theory that would answer the question why patients suffer from this disorder is still lacking.

Reports of (hypo)manic episodes after experiencing COVID-19 are starting to emerge in scientific literature [24,25,26,27]. As previously stated, manic episodes are the key element of BD. Therefore, we believe that in the light of new scientific reports regarding cases of mania, as well as constant, new discoveries related to COVID-19, we should analyze these type of psychiatric disorders, determine its association with SARS-CoV-2 and consider a possibility of undiagnosed or misdiagnosed BD among patients with psychiatric sequel of COVID-19 [28,29]. As BD is a neuroprogressive disorder, it is important to detect it at the earliest possible stage or be prepared for even the possibility of its development. That would allow for quick implementation of appropriate treatment to minimize the effects of the disease and maximize patients’ years of life with a quality similar to that of healthy people.

The aim of this study is to try, based on the available literature, to answer the question of whether BD may be one of the neuropsychiatric manifestations of COVID-19. For this purpose, we will collect data on the prevalence of this type of affective disorder among patients who have been infected with SARS-CoV-2-CoV. We will also compare the disorders and changes caused by this virus and how it affects the homeostasis of the organism with the disorders and changes present in patients with bipolar disorder, including immunological, endocrine, biochemical, and structural. Additionally, we will describe the disorders found by us in the context of the most common hypotheses and risk factors for the development of BD. Gathered information will allow for a better estimation of the long-term neuropsychiatric effects of COVID-19 and the development of better strategies for diagnosis and treatment. In addition, it might serve as a source of knowledge to better explain the etiopathogenesis of BD and help patients who already suffer from this disease.

## 2. Materials and Methods

A literature search was conducted in PubMed, Scopus, and Google Scholar databases. We included clinical studies, reviews, meta-analyzes and case studies regarding manic symptoms in COVID-19 (+) patients. The search strategy consisted of the following keywords: ‘mania’, ‘manic episode’, ‘COVID-19’, ‘SARS-CoV-2’, ‘post-COVID’, ‘bipolar’, ‘psychiatric sequel’ as well as combinations of these terms. We then reviewed the abstracts and excluded all articles in which manic episode or psychiatric events in general were described in the context of the social consequences of a pandemic, stress, fear or economic problems, in order to be able to analyze only the direct impact of SARS-CoV-2 on the body. Relevant studies were then included with the intention of covering the widest possible spectrum of manic symptoms in patients previously infected with SARS-CoV-2. We conducted additional manual searches of the references of the related articles in order to gather information about the relevant supporting literature.

## 3. Results

From the available literature, we included in our analysis a total of 22 publications describing cases of patients who experienced a (hypo)manic episode correlated with COVID-19. In total, there were 36 patients. The vast majority were male, patients were of different ages and nationalities, and the greater part had never been treated for mental disorders and had no family history of such disorders. Twelve patients were previously diagnosed with psychiatric illness, six of which experienced manic episode in the past [30]. The severity of COVID-19 was different among them, most often it was mild or moderate infection, and the manic symptoms developed at varying time intervals from the onset of COVID-19 symptoms. In three patients, SARS-CoV-2 infection was not confirmed by a formal test, but only on the basis of the clinical picture and characteristic imaging changes in the lungs [27,31,32]. The most common symptoms of mania among them were: insomnia, elevated mood, agitation, irritability, delusions, grandiosity and talkativeness. Acute manic episode was diagnosed in 26 patients [24,25,26,27,30,31,32,33,34,35,36,37,38,39,40], hypomanic episode in 3 [32], steroid-induced mania in 4 [41,42,43,44], delirious mania in 2 [45,46], and a full diagnosis of BD was given in 1 patient [47]. The details of the patients are described in Table 1.

To understand the possible causes of hypo(mania) and potential BD in COVID-19 patients, we collected a number of publications from which we included and analyzed 108 publications regarding biomarkers and frequent changes in BD patients as well as 70 publications regarding disorders connected to SARS-CoV-2 infection and compared them. Our analysis allowed us to divide those changes and disorders into: inflammatory factors, the effect of medications, disorders of the hypo-thalamic-pituitary-adrenal axis (HPA), sleep disorders, growth factor disorders, kynurenine pathway activation, structural changes in the central nervous system (CNS), disturbances in neurotransmission and oxidative stress. The similarities and differences between abovementioned factors found by us among BD and SARS-CoV-2 patients are described below.

### 3.1. Inflammatory Factors

Pro-inflammatory cytokines include interleukin-1β (IL-1β), interleukin-2 (IL-2), interleukin-6 (IL-6), tumor necrosis factor α (TNF-α), and interferon gamma (IFN-γ), and are mainly produced by Th1 lymphocytes and M1 monocytes/macrophages, and microglia. Th1 cells are mainly responsible for the immune response to infections with intracellular bacteria and viruses, and M1 macrophages recruit and activate lymphocytes at the site of inflammation. The anti-inflammatory cytokines are interleukin-4 (IL-4), interleukin-5 (IL-5), interleukin-10 (IL-10) and interleukin-13 (IL-13) and are mainly produced by Th2 lymphocytes, Treg, M2 monocytes/macrophages and astroglia. Th2 cells are responsible for the humoral response, are active in allergic reactions and in intestinal nematode infections. M2 macrophages are involved in wound healing, tissue repair and remodeling after inflammation [48,49,50]. Considering the influence of inflammatory factors on the development of neuropsychiatric disorders, the most important seems to be the influence of cytokines on the CNS. Cytokines can be produced in the brain by astrocytes and microglia [51,52], or penetrate into it from the periphery through leaky blood–brain barrier (BBB), active transport, activation of endothelial cells and binding to cytokine receptors, especially in the case of increased cytokines level in the body [53]. Their level is elevated especially during the so-called cytokine storm which is a severe immune reaction involving a rapid and excessive release of cytokines in the body that can cause damage and dysfunction of many organs, including the CNS [54]. Inflammatory mediators, such as TNF-α, affect the endothelial cells that make up the blood–brain barrier and increase its permeability [55]. Cytokines affect astrocytes and microglia and cause an increase in production of subsequent cytokines by those cells, which initiates a cascade leading to the development of CNS inflammation [51,52,56]. Pro-inflammatory cytokines destabilize brain function and make it more vulnerable to stress. In addition, they influence neuroplasticity, hippocampal neurogenesis, and act as neuromodulators by disrupting neurochemical and neuroendocrine pathways [57,58,59]. As already mentioned, excessive and uncontrolled cytokine release may significantly influence the development of neuropsychiatric disorders including, e.g., post-COVID-depression [7,60]. According to some hypotheses of BD etiopathogenesis, cytokine disorders might also be one of the contributing factors [61,62].

During mania, the levels of cytokines such as IL-1, IL-2, IL-4, IL-6 and TNF-α are increased [61,62]. As in the case of other psychiatric diseases with an inflammatory component, it is difficult to determine whether the increase in these markers is the result or the cause of the BD. Berk et al. (2011) however, showed that in the early stages of the disorder, the anti-inflammatory cytokine IL-10 is elevated and its level decreases with the progression of BD, while pro-inflammatory IL-6 and TNF-α are elevated at the onset of the disease and persist over time which may suggest that persistent inflammation and depletion of anti-inflammatory mechanisms causes the progression of BD [63]. Higher level of inflammatory markers, namely C reactive protein (CRP), TNF-α and IL-6 in depressed patients was also associated with a higher risk of developing manic symptoms compared to depressed patients in whom the level of these markers was lower [64]. It is also worth mentioning that the peripheral inflammatory markers in patients with BD are different from those in MDD patients. In BD depression, IL-6 is the predominant cytokine, in euthymic state it is IL-4, and in manic episode it is IL-2, IL-4, IL-6, while patients with unipolar depression have mainly elevated TNF-α, IL-6 and soluble IL-2 receptor (sIL-2R) [65]. Differences in the cytokine profile between BD and MDD were also confirmed by using machine learning—higher levels of IL-10, IL-4 and thiobarbituric acid reactive substances (TBARS) proved to be good at distinguishing between bipolar disorder from unipolar depression [66]. Ortiz-Domínguez et al. (2007) describe a significant increase in IL-4 levels and a decrease in IL-6 and IL-1β in patients with BD in the manic phase and a reverse trend in the depressive phase, as well as a significantly higher level of TNF-α and decreased level of IL-2 in both phases [67]. However, not all results regarding the differences in the cytokine profile depending on the phase of BD are consistent and the opposite direction of changes in cytokine levels has also been reported. Kim et al. (2007) described increased levels of IL-6 and TNF-α and significantly decreased level of IL-4 in BD patients during the manic episode compared to healthy people, and also no differences in IL-2 levels [61]. In a large meta-analysis of 30 studies, Modabbernia et al. (2013) observed in BD patients an increase in cytokines such as IL-4, IL-6, IL-10, TNF-α and the cytokine receptors: soluble TNF receptor 1 (sTNFR1), sIL-2R and soluble IL-6 receptor (sIL-6R). The levels of IL-2 and interleukin-8 (IL-8) did not differ between patients and control groups. Additionally, some changes in cytokine levels were dependent on the BD phase, i.e., TNF-α, sTNFR1, sIL-2R, IL-6, and some, e.g., IL-4 and IL-10, were independent of it [68]. Although the results of the studies are inconclusive, in most cases they show a general Th1 cells activation in the course of BD, and some of the results additionally indicate the participation of Th2 cell activation and an increase in anti-inflammatory cytokines during mania, as opposed to an increase in pro-inflammatory cytokines during a depressive episode [65,66,67]. This coincides with the spring/summer peak periods of mania when Th2-mediated allergies increase, and the depressive episode periods, i.e., in winter and autumn, when the number of infections increases and the Th1 response is triggered [69].

Cytokine storm has been widely described in relation to COVID-19 [70,71]. Among COVID-19 patients, the most common disorders of cytokines and chemokines are increased levels of IL-1β, IL-2, IL-6, interleukin-7 (IL-7), IL-10, TNF-α, CRP, chemokine ligand 2 (CCL2), as well as an increase or decrease in the concentration of IFN-γ [7,70,72]. During the first 2 weeks of symptoms of the acute phase of COVID-19 infection, T cells activate Th1 which leads to further activation and polarization of monocytes and the production of pro-inflammatory cytokines such as IL-2, IL-6, TNF-α, etc. [73]. However, in the course of severe infection, Th2 activation occurs, which is associated with a worse prognosis [74,75]. The direction of these changes is therefore similar to the changes in the Th1/Th2 ratio in the course of BD described in the literature. In addition, in the case of people with a severe course of COVID-19, a greater Th17 response and related interleukin-17 (IL-17), interleukin-21 (IL-21), interleukin-22 (IL-22) and granulocyte-macrophage colony-stimulating factor (GM-CSF) were also noted [76]. The scientific literature describes the inhibitory effect of IL-17 on the development of Th1 [77], and its inducing effect on the development of the Th2 response [78]. It seems that the cytokine profile among COVID-19 patients is similar to that in patients with BD and may create favorable conditions for the development of this disorder. However, among the gathered COVID-19 patients with mania symptoms, only 16 out of 36 showed an increase in markers of inflammation and cytokines [24,25,27,30,40,42,43,45,47,79], which may be attributed to the small number of severely ill COVID-19 patients among the collected cases [80].

In summary, in BD, as in MDD, inflammatory factors play a significant role in its etiopathogenesis. Their entry into the CNS through the BBB and their very production in the brain by astrocytes and glial cells affects the development of neuroinflammation and destabilization of brain functions. They also increase susceptibility to stress, influence neuroplasticity, hippocampal neurogenesis, and disrupt neurochemical and neuroendocrine pathways [51,52,57,58,59]. The cytokine disorders appearing in BD include an increase in the level of pro-inflammatory cytokines and activation of the Th1 response, but in the case of a manic episode, attention is drawn also to the induction of Th2 responses and an increase in anti-inflammatory cytokines such as IL-4, IL-5 and IL-10 [63,65,67,81]. A very similar cytokine profile can be seen in COVID-19 patients. During cytokine storm there is an increase in pro-inflammatory cytokines, and in later stages of cytokine storm a Th2 activation can be seen [70,72,74,75]. However, it cannot be ruled out that the increase in the level of anti-inflammatory cytokines is a compensatory mechanism in response to significantly increased levels of pro-inflammatory cytokines and other factors, such as, e.g., oxidative stress [68]. Despite this possibility, the seasonality of manic episodes occurring in the spring-summer periods, when allergic reactions mediated by Th2 response intensify, speaks for the influence of anti-inflammatory factors on the development of mania [69].

### 3.2. Effect of Pharmacotherapy

A number of pharmacological agents can contribute to a manic episode. Substances for which convincing evidence has been gathered to induce mania include levodopa, corticosteroids, anabolic steroids and amphetamine. Others for which the evidence is weaker include chloroquine, baclofen, thyroxine, isoniazid, alprazolam and captopril. Tricyclic antidepressants and monoamine oxidase inhibitors can induce a manic episode in people with pre-existing BD [82,83]. For the sake of clarity, we will focus here on a group of drugs that are most often used in severe infections with difficult to control systemic inflammation-corticosteroids (CS).

A case study of manic episodes following the use of corticosteroids showed that the smallest dose of CS that can cause such an effect is 20 mg per prednisone [84]. A greater risk of developing a manic episode occurs when the dose of 40 mg of prednisone (or equivalent) per day is exceeded, especially in the case of short treatment with medium or high doses [85,86]. In the case of a dose greater than 80 mg of prednisone, 18,6% of patients experience adverse effects in the form of neuropsychiatric disorders [87]. The overall incidence of steroid-induced mania varies from study to study. Lewis and Smith rated that psychiatric syndromes occur in approximately 5% of patients treated with steroids, and that almost 30% of it are manic episodes [88]. A similar value, i.e., 26%, was observed among ophthalmological patients [89]. In a more recent study, Nishimura et al. (2008) observed an incidence of 6.5% among patients receiving CS for the treatment of systemic lupus erythematosus (SLE) [90]. In a study evaluating the effects of long-term therapy with low-dose corticosteroids, there was a 60% incidence of affective disorders among patients, with depressive disorder being more common than mania. The authors of the study hypothesized that the method of steroid administration influences the type of affective disorder, and that short-term high-dose steroid therapy predisposes to mania, in contrast to long-term low-dose therapy, which is more likely to cause depression as has been reported in relation to Cushing’s syndrome [91,92]. Affective disorders appear early after initiation of CS therapy, usually within the first week [84,87]. The manic episode usually resolves within 14–41 days after steroid withdrawal and initiation of pharmacotherapy [88,93], although there are reports of mania lasting many months despite steroid discontinuation [87,94,95]. In euthymic bipolar patients, the administration of corticosteroids may lead to the manic switch [96]. The mechanism of steroid-induced mania remains unclear. Some hypotheses focus on the effects of exogenous steroids on the HPA axis. The supply of steroids inhibits the HPA axis and reduces the production of endogenous cortisol. Synthetic steroids have a greater affinity for glucocorticoid receptors than mineralocorticoid receptors, so in the case of an endogenous cortisol deficiency, there is an increased activation of the glucocorticoid receptor, which may result in cognitive and emotional disorders [42,95]. In addition, high doses of corticosteroids may interfere with dendritic branching, neurogenesis, glucose availability in the hippocampus, and the production of neurotrophic factors [95]. Corticosteroids also increase the activity of tyrosine hydroxylase, and thus the production of noradrenaline (NA) and dopamine (DA), which are involved in mania [97].

The use of corticosteroids is common in COVID-19 therapy. The National Institute of Health (NIH) in its guidelines for the treatment of hospitalized COVID-19 patients recommends 6mg of dexamethasone (equivalent to 40 mg prednisone) daily for 10 days or until discharge from the hospital [98]. There is no evidence in the scientific literature that the other drugs included in the NIH guidelines, such as remdesivir, baricitinib, heparin, tofacitinib, tocilizumab, and sarilumab, could cause manic episodes. On the other hand, recommended dose and the cycle length of dexamethasone treatment coincide with the average dose of corticosteroids and the short cycle capable of causing a manic episode. Of the cases we collected, 14 out of 36 patients received treatment with corticosteroids and they were implemented at moderate doses, as far as we know [Table 1]. One of the reported cases of mania after administration of 30 mg of prednisone may suggest a decrease in corticosteroid tolerance caused by SARS-CoV-2 infection in a patient chronically taking high doses of steroids for SLE [42].

In summary, the side effects of corticosteroids include neuropsychiatric disorders, among which mania is one of the most common [82,83,99]. The average dose and cycle length after which such disorders may appear is a short cycle of minimum 40 mg prednisone, which corresponds to the recommendations for administering steroids to patients with COVID-19 issued by NIH [85,86,98]. However, among the cases collected by us, only 38.8% of patients received treatment with corticosteroids and this may indicate a different mechanism of mania development among COVID-19 patients than just the effect of pharmacotherapy [Table 1]. In one case, a patient stopped tolerating steroids and developed a manic episode only after being infected with SARS-CoV-2 [42]. The influence of steroids cannot be ruled out here, but this case may suggest a cumulative effect of pharmacotherapy and viral infection and all the changes in homeostasis that follow.

### 3.3. Disorders of the Hypothalamic-Pituitary-Adrenal Axis

The hypothalamic-pituitary-adrenal axis (HPA) is one of the body’s neuroendocrine systems involved in the stress response. Its basic product is cortisol. This hormone exerts its effects through the glucocorticoid receptor (GR) and the mineralocorticoid receptor (MR) which are present, inter alia, in the central nervous system. MR has a higher affinity for cortisol and is found mainly in the hippocampus, prefrontal cortex, and the amygdala (limbic system). GR has a lower affinity and is distributed in most structures of the brain. It can be found in oligodendrocytes, astroglia and microglia [100,101]. As we have already mentioned, the effects of corticosteroids on the brain are remarkably wide—they affect neuronal differentiation, integrity and growth of neurons, as well as synaptic and dendritic plasticity [86,102]. The basic and slightly elevated level of cortisol enhances the above-mentioned processes, and a significantly increased level has the opposite effect [103]. The effects of corticosteroids on cellular processes result in their effects on bodily functions such as energy regulation, learning and memory, visual information processing, movement control, decision making, and reward-oriented behavior [102], as well as remodeling of the hippocampus, amygdala and prefrontal cortex, which together result in changes in human behavior and psyche [101].

Among patients suffering from BD, a number of deviations in the activity of the HPA axis have been described, but they vary from study to study and there is still no consensus on this matter. Girshkin et al. (2014) in their meta-analysis reported a slight increase in cortisol in euthymic or depressed patients with BD, but did not find any difference between patients in a manic episode and a healthy control group [104]. On the other hand, a large meta-analysis indicated that patients with BD suffer from HPA axis hyperactivation, especially in the manic phase, but it also persists during remission. The authors also pointed out that prolonged hypercortisolemia may lead to the recurrence of affective or psychotic disorders in stable patients with BD. Additionally, disturbances in GR function and a reduction in GR mRNA levels in the hippocampus (HPC) and amygdala (AMG) have been detected in patients with BD which may be the result of prolonged hyperactivation of the HPA axis [105]. Becking et al. (2015) conducted a study in which a large diurnal decrease in cortisol combined with an elevated CRP level was associated with the occurrence of mania and hypomania in men with previously diagnosed depression [106]. A recent study by Markopoulou et al. (2021) however, indicated that in patients with treatment resistant BD the total cortisol is lower than in the control group, which may indicate a shift from hyperactivation to hypoactivation of the HPA axis during the course of the disease [107,108].

In COVID-19 and the systemic inflammation it causes, a response from the HPA axis is natural. However, research in this area is inconclusive. One study showed higher cortisol level among COVID-19 (+) patients compared to COVID-19 (−) patients, and higher mortality which corresponding to increasing cortisol levels [109]. An increase in cortisol level, proportional to the severity of the course and associated with a higher risk of death was also reported in other studies [110,111]. However, some researches point in the opposite direction and describe hypocortisolemia in COVID-19 patients, especially in those with a severe course of infection [112,113]. In the context of COVID-19, there are also cases of newly diagnosed adrenal insufficiency (AI) confirmed by the adrenocorticotropic hormone (ACTH) stimulation test, and their incidence, depending on the study, reaches up to 64% [114,115,116]. Critical illness-related corticosteroid insufficiency (CIRCI) is a known phenomenon, and its occurrence is associated with a decrease in cortisol transport and a reduction in its breakdown, insufficient GR-α activity, inhibition of ACTH secretion and steroidogenesis [117,118]. In the case of COVID-19, the AI hypotheses include (1) direct effect of SARS-CoV-2 virus on the pituitary and hypothalamus, which express angiotensin-converting enzyme 2 (ACE2) receptors (2) direct effect on the adrenal glands (3) similarity of the viral amino acid sequence to the ACTH amino acid sequence and immune cross-reaction (4) reduction in ACTH release induced by TNF-α (5) dysfunction of the HPA axis caused by cytokine storm (6) critical condition and the resulting CIRCI (7) reduction in adrenal venous drainage by artery dilating effect of ACTH (8) effect of exogenous corticosteroids [116,118,119]. To date, however, few cases of AI have been reported in COVID-19 patients compared to studies showing an increase in cortisol in infected patients. It is also worth noting that reports of histological and morphological changes in the adrenal glands in SARS-CoV-2 infected patients mainly come from the autopsy of patients with comorbidities and in whom severe disease led to multi-organ failure [120,121].

In summary, infection with SARS-CoV-2 leads to disturbances in the HPA axis but the direction of those disturbances is not consistent across studies—same as for BD [109,110,112]. In relation to COVID-19, a large proportion of research on the HPA axis reports hypercortisolemia, but there are also references to hypocortisolemia and AI [116,118,119]. High cortisol levels in COVID-19 patients could possibly be associated with its subsequent decline and large diurnal fluctuations in its level, which could cause manic symptoms [106]. However, to prove this, in the future, multiple measurements of cortisol level in COVID-19 patients and establishing the daily profile and fluctuations in its level would be needed. However, based on the data so far, it is not clear whether disturbances in the stress response may be the cause of mania or BD in patients with COVID-19. The stress response system is a complex mechanism and its involvement in the development of these disorders cannot be completely excluded, especially in combination with other disorders in the body caused by severe SARS-CoV-2 infection.

### 3.4. Sleep Disturbances

A reduced need for sleep is one of the key symptoms of (hypo)mania. It may also be a prodromal symptom of a manic state and appear many weeks before development of other characteristic features of the condition. On average, 77% of people with BD indicate sleep disturbance as a symptom that emerges before the development of a full episode of mania [122]. Malkoff-Schwartz et al. (1998, 2000) observed that patients with BD indicated more social rhythm disturbances involving changes in sleep or waking hours during the 8 and 20 weeks preceding full manic episode [123,124]. Sleep deprivation is also used in animal models to generate manic symptoms. After sleep deprivation, mice temporarily exhibit subsequent insomnia, hyperactivity, aggressiveness, and hypersexuality [125].

Sleep deprivation can also cause a manic episode in previously healthy individuals. New cases of mania have been described in the context of several days of significant reductions in sleep duration or chronic sleep reduction. Patients returned to normal functioning shortly after returning to an appropriate sleep duration [126,127]. However, sleep deprivation-induced mania can last much longer. Wehr et al. (1987) described the cases of 4 patients with different psychiatric history (BD, MDD or no previous diagnosis) who developed symptoms of (hypo)mania after sleep deprivation of various duration (from 1 night to several nights in a row). These symptoms lasted for up to several months [128]. The ability to induce mania in response to sleep deprivation has been used as an attempt to treat depression. Colombo et al. (1999) noticed that a total of over 10% of depressed patients treated with this method developed (hypo)mania during the study [129]. Similar results were seen in a study that compared the effects of sleep deprivation in patients with BD and MDD. In the bipolar depression group, 80% of patients achieved an average of 50% improvement in the Hamilton Depression Rating Scale (HDRS) score after sleep deprivation [130]. In the MDD group, a temporary improvement by 30% in 53% of patients was observed and it was accompanied by a nocturnal rise in cortisol level which lasted longer in patients who showed improvement in depressive symptoms [131]. Additionally, in a study by Song et al. (2015), the authors observed a significant increase in cortisol level and a significant increase in the mania rating scale score in a sleep-deprived group compared to the control group. In addition, according to authors, an increase in cortisol level above 500 nmol/L may be used as a predictor of a manic episode with a sensitivity of 74% and a specificity of 100% [132]. Currently, there is no generally accepted hypothesis explaining the phenomenon of mania induction and the antidepressant effect of sleep deprivation. However, the abovementioned increase in cortisol after a sleepless night could cause a short-term activation of the HPA axis with subsequent normalization of its activity, improvement of the negative feedback response to the hypothalamus and reduction in corticotropin-releasing hormone (CRH) [131,133]. Moreover, an increase in cortisol may induce the production of DA, one of the key neurotransmitters in the development of mania (discussed later) [133,134] and the increase in brain-derived neurotrophic factor (BDNF) level. One of the effects of BDNF is further stimulation of the release of DA, as well as serotonin (5-HT), γ-aminobutyric acid (GABA) and glutamate [135,136].

Sleep disturbances are often part of the so-called “Long-COVID”. In a large study assessing 6-month neurological and psychiatric outcomes, involving nearly 240,000 patients who underwent COVID-19, it was observed that first-time insomnia occurs on average in 2.5% of patients who have had COVID-19, and the incidence increases with the severity of the infection. In people who required hospitalization in the intensive care unit it was 4.24%. In a similar but smaller study involving 1,733 people, sleep disturbances were reported in 26% of the subjects [137]. However, the outcomes of the study by van den Ende et al. (2021) oppose the abovementioned results. The authors surveyed COVID-19 patients during their hospital stay and discovered that COVID-19 patients reported a complete lack of sleep five times more often than other hospital patients, but the average total sleep time differed between the groups by only 15 min. However, the authors pointed out a different distribution of sleep time among the two groups, namely, in COVID-19 patients, it was often 0 h of sleep or more than 13 h of sleep, while in the COVID-19 negative group, in most cases, this time oscillated around the average of 6 h [138].

Summing up, sleep disturbances are one of the features of a manic episode, which may also be a prodromal symptom in patients already diagnosed with BD [122]. It is also one of the risk factors for mania, and healthy individuals who experience sleep deprivation episodes, sometimes even as short as 1 day, may develop mania symptoms of varying duration [126,127,132]. Additionally, among patients with MDD or BD, sleep deprivation is associated with a subsequent improvement in mood or the development of a (hypomanic) episode [128,129,130,131]. In relation to COVID-19, sleep disturbances and insomnia are reported by patients and described by the authors of the studies relatively often [3,137,138]. This is quite a unique feature of COVID-19 and is strongly associated with the development of mania and should therefore be taken into account in the overall assessment of the risk of developing BD after COVID-19. Even if one sleepless night may result in mania, it should be remembered that one of its symptoms may be further insomnia, and thus a self-perpetuating cycle is set in and the manic episode may be protracted.

### 3.5. Brain-Derived Neurotrophic Factor

Brain-derived neurotrophic factor belongs to the family of neuronal growth factors and is also the most common neurotrophin in the central nervous system. It acts through the TrkB receptor and is responsible for the differentiation, growth and survival of dopaminergic, cholinergic and serotonergic neurons in the central nervous system [135,139]. It also has neuroprotective abilities, contributes to the maintenance of synaptic plasticity and long-term memory enhancement. BDNF molecule crosses the BBB and can therefore be detected in the blood serum [140].

BDNF secretion disturbances are frequently described in the context of affective disorders. In patients with MDD, decreased serum level and its recovery to normal values after treatment with antidepressants have been reported [141,142]. This topic is also discussed in relation to BD, although the research results are not unequivocal. Meta-analyzes have shown that among patients with BD there is a decrease in BDNF levels in the manic and depressive phase, and during euthymia it is comparable to the control group [143,144,145]. Post-mortem studies also showed decreased level of BDNF in the brains of BD patients [146]. However, some studies also indicated an increased level of BDNF among patients with newly diagnosed BD—up to 22% compared to the control group. It was also noted that the level of BDNF positively correlated with the duration of the disease, which is in contrast to the results of other studies [147], which showed that BDNF is reduced only in patients with advanced stage of BD, and its levels decline with the duration of the disease [148]. Lin et al. (2016) also did not seem to confirm the reduction in BDNF in manic patients. Furthermore, 4 weeks of treatment with mood stabilizers did not change the BDNF level [149]. In a work by Tsai (2004), the author hypothesized that mania was caused by BDNF overactivity. According to the research, factors that can induce mania also increase BDNF, e.g., administration of amphetamines, hyperthyroidism or antidepressants [150]. This theory is supported by research on sleep deprivation. Rapid eye movement (REM) sleep disturbances can relieve symptoms of depression or cause mania. At the same time, after REM sleep deprivation, there is a sudden and short elevation in the BDNF level [129,130,151,152,153].

Among COVID-19 patients, the described changes in the level of BDNF mainly concern its decreased level. In a study involving 64 COVID-19 (+) patients there was a significant reduction in BDNF level among patients as compared to the control group. This difference was greatest in a group with the neurological manifestation of COVID-19, and smaller in patients with only fever and dyspnea [154]. A decrease in BDNF level in COVID-19 patients depending on their condition has also been observed in other studies and it was negatively correlated with the severity of the patients’ condition, and its normalization followed the improvement of the clinical condition [155,156]. The use of ACE2 receptors by the SARS-CoV-2 virus and their subsequent downregulation may be an indirect cause of the decrease in BDNF levels. The ACE2 product, angiotensin 1–7 (Ang (1–7)), causes the production of the Mas protein and hence the formation of BDNF. ACE2 deficiency will therefore result in reduced BDNF production [157,158,159]. However, there is also evidence of elevated BDNF levels in COVID-19. In an interesting pilot study involving a small group of young adults, the level of BDNF after being infected with SARS-CoV-2 was examined. The results showed that its level was higher in patients as compared to the healthy control group, but only in girls with symptomatic infection and in those girls who later developed the symptoms of the “long-COVID” [160].

Most studies seem to confirm that COVID-19 infection causes a decrease in BDNF levels [154,155,156]. These results, however, are not consistent in all cases and there are studies that do not show a decrease in BDNF, or even its increase among patients who developed “long-COVID” [160]. The decrease in BDNF is also noticeable among patients with BD, although the opposite results can be found in this case as well [144,145,147]. Such a picture may correspond to the hypothesis that there is an overall decrease in BDNF during the course and progression of BD, but manic episodes may be triggered by a temporary and rapid increase in BDNF secretion [150]. In such case, it is possible that the stress itself caused by a viral infection or the rapid normalization of neurotrophin secretion during treatment of SARS-CoV-2 could be a trigger for the development of a manic episode. However, this hypothesis requires further and more in-depth research.

### 3.6. Kynurenine Pathway

The kynurenine pathway is one of the tryptophan (TRP) metabolism pathways, which is strongly associated with neurodegenerative diseases, inflammation and affective disorders—especially MDD. In this pathway, tryptophan is redirected to the production of kynurenine (KYN) and its metabolites [161]. There are two enzymes involved in this process—indoleamine 2,3-dioxygenase (IDO) and tryptophan 2,3-dioxygenase (TDO) [162]. Kynurenine pathway two end products are acetyl coenzyme A (Acetyl-CoA) and nicotinamide adenine dinucleotide (NAD). Their formation is accompanied by the production of large amounts of intermediate metabolites such as anthranilic acid (AA), kynurenic acid (KYNA), 3-hydroxykynurenine (3-HK), 3-hydroxyanthranilic acid (3-HAA), quinolinic acid (QA) and picolinic acid (PA) [163]. Among them, KYNA is a N-methyl-D-aspartate (NMDA) receptor antagonist, an inhibitor of the α7 nicotinic acetylcholine receptor and a stimulant of the neprilysin (NEP) gene expression, thanks to which it exhibits neuroprotective properties [164]. On the other hand, QA is an NMDA receptor agonist, a precursor of NAD and it has been associated with neurodegeneration and excitotoxicity [165]. 3-HK also shows a strong toxicity towards neuronal cells. It causes their apoptosis and death, inter alia, by generating oxidative stress [166]. In the brain, cells such as microglia and astrocytes express IDO, but the main metabolite of the kynurenine pathway is different for them. For microglia it is QA, and for astrocytes KYNA [167,168]. Activation of IDO can occur through stimulation with pro-inflammatory cytokines such as TNF-α, IL-6 and the most potent of its activators—IFN-γ [167]. Various cytokines also lead to the activation of a specific pathway of KYN metabolism—Th1 cytokines activate microglial IDO and will increase QA production, while Th2 cytokines will deactivate microglial IDO and redirect KYN metabolism towards astroglial KYNA production [65].

Changes in TRP metabolism and the accumulation of neurotoxic metabolites is discussed not only in MDD but also in BD [17,169]. In the group of bipolar patients who committed suicide, NAD—the end product of the kynurenine pathway, was significantly elevated in their anterior cingulate cortex (ACC) compared to BD patients who died of different reasons. Significantly elevated KYNA, PA and the KYN/TRP ratio in the cerebrospinal fluid has also been found in the abovementioned group [170,171]. During manic phase, the activity of IDO (KYN/TRP) and the production of 3-HK and QA are positively correlated with the level of TNF-α, and during depressive phase, patients have a lower KYNA/3-HK [172]. It was observed that during different phases of BD there was an equal degree of activation of kynurenine aminotransferase (KAT)—which catalyzes the conversion of KYN to KYNA, and activation of kynureninase (KYNU)—which catalyzes the conversion of 3-HK to 3-HAA. However, in manic group there was a stronger conversion of KYN to 3-HK [173]. Meta-analysis by Bartoli et al. (2021) showed the lowest concentration of TRP in the blood serum of patients in manic episode and the lowest concentration of KYNA among patients in depressive episode [174]. Low KYNA concentration is also associated with the deterioration of cognitive functions among BD patients during a depressive episode, but surprisingly has no effect on cognitive functions among the manic group [175].

TRP metabolism disorders are one of the few serious metabolic disorders found in COVID-19 (+) patients. The activation of the kynurenine pathway in these patients is indicated by increased concentrations, e.g., KYN, KYNA and PA which positively correlate with the IL-6 serum level. These disorders are so specific that the level of TRP below 105 μM and KYN above 5.3 μM has 95% of AUC in the differentiating of the COVID (−) and COVID (+) groups [176]. Many studies have also shown increase in the KYN/TRP ratio and its positive correlation with the severity of COVID-19 [177,178,179,180]. Other metabolites of kynurenic acid which are elevated in SARS-CoV-2-infected patients include 3-HK, PA, AA and 3-HAA [177,179,180]. Slightly different results have been observed by Almulla et al. (2022). Similar to other researchers, they noticed an increase in the KYN/TRP ratio but it was not accompanied by a significant increase in KYNA [178]. All the above-described changes are indicative of activation of the kynurenine pathway among COVID-19 patients and appear to shift towards Th1 mediated activation of microglial IDO. Interestingly, the Th2 anti-inflammatory cytokine—IL-4, which is released during COVID-19 cytokine storm, in addition to its ability to activate astrocytes, appears to have the ability to potentiate IFN-γ-mediated microglial IDO activation [181]. When discussing disturbances in tryptophan metabolism, it should also be added that the ACE2 receptor is highly expressed in the intestines, where it is essential for the expression of the neutral amino acid transporter in the intestinal lumen—B0AT1. Thus, downregulation of ACE2 and lower B0AT1 expression results in decreased tryptophan absorption from the gut [182,183].

In summary, the activation of the kynurenine pathway in BD causes the accumulation of substances with mainly neurotoxic, excitotoxic and pro-inflammatory properties, and to a lesser extent and depending on the activation route—neuroprotective [65,164,165]. In patients with mania, the activity of IDO seems to be shifted towards the activation of its microglial variant, and the level of TNF-α correlates with it and with the production of 3-HK and QA [172,173]. Similar changes occur during COVID-19. Strong activation of the kynurenine pathway and a shift in IDO activation towards the microglial side, induced by Th1 cytokines, are also observed among the infected patients. Additional activation of microglial IDO is ensured by the secretion of the cytokine IL-4 during a cytokine storm [177,179,181]. The convergence of these changes may indicate the role of kynurenine pathway activation in the course of COVID-19 and the development of affective disorders (including mania and BD). Due to the large share of kynurenine pathway in the development of affective disorders, more detailed research is needed in the future.

### 3.7. Structural Changes in the CNS

Structures such as the prefrontal (PFC) and frontal cortex (FC), the striatum and the limbic system (LS) are involved in shaping emotions, psychomotor drive, behavior, control sex drive, hunger and thirst, synchronize sleep patterns and are critical to decision making and memory. It can be said that all the above makes up a human personality. During affective episodes in patients with BD, most of these components suddenly change, so naturally a search for structural changes in the regions of the brain related to the formation of personality began. Thanks to modern imaging methods, it is known today that the CNS image in patients affected by BD (and many other psychiatric diseases) differs from that in healthy people [184].

The changes that are clearly visible in the CNS imaging in bipolar patients are morphometric changes such as an increase in the lateral and third ventricle volume and a reduction in the size of the HPC, AMG, fusiform gyrus (FG) cingulate gyrus (CG), striatum (STR) and gray matter (GM) volume in many regions of the brain, including prefrontal and frontal [185,186,187,188]. Research by teams belonging to the ENIGMA Bipolar Disorder Working Group showed a reduction in the thickness of the cerebral cortex in the frontal, temporal and parietal regions. These changes depend on the duration of the disease and the number of previous affective episodes, in particular (hypo)manic [189,190]. Changes such as a reduction in HPC volume and density of the cerebellar GM, as well as an increase in ventricular volume also progress with each subsequent affective episode and disease duration [186,187]. According to a large review study by Teixeira et al. (2019) the most frequently recurring changes in the cortical GM of bipolar patients are the reduction in its volume in the areas of the ACC, the left superior temporal (STG), parahippocampal (pHPC), FC and PFC regions [191].

Another significant and frequently described abnormalities seen on magnetic resonance imaging (MRI) or computed tomography (CT) images are hyperintense changes in the deep white matter (WMH) of bipolar patients’ brains. The processes associated with WMH include demyelination, gliosis and axonal degeneration, which in practice means disturbances in the connectivity between individual parts of the brain [192,193]. The chance of finding WMH in the brain of BD patients is 2.5 times higher than in healthy subjects [194]. Additionally, the amount of WMH found in their brain positively correlates with the number of previous manic episodes. These changes also appear more frequently among healthy family members of BD patients than among those without the family history of this disorder [195]. Diffusion tensor imaging (DTI) studies show that white matter abnormalities occur in the prefrontal, frontotemporal and cingulate regions, and indicate loss of projection, associative and commissural fibers [196,197,198]. The prefrontal lobes are key to the planning and evaluation of goal-oriented behavior and comprise the regions involved in working memory. Thus, disorders in the white matter and connections of the prefrontal lobes with the cortical and subcortical regions can manifest themselves with symptoms such as incorrect assessment of behavior and poor decision making. In the frontal lobes, there are centers involved in emotional control, drive, judgment, and sex drive. These components are also disturbed during an affective episode in patients with BD, and WMH and impaired neural circuits may be the cause [196,199]. Disturbances of the connectivity between the medial and lateral prefrontal regions may result in a reduction in the volume of the ventrolateral prefrontal cortex with its increased activation as a result of compensation and hyperactivation of long-cortical neurocircuits with subsequent stimulation of regions such as the HPC, AMG and thalamus. It may also cause an overactivity of the left ventral striatal-ventrolateral and orbitofrontal cortical circuitry, related to the reward system [199,200].

Regarding COVID-19, changes in the brain image are also described, including those with a similar location and nature as in patients with BD. In a study by Huang et al. (2022) using DTI, it was observed that one year after COVID-19, patients still had abnormalities in white matter (WM), especially in the fibers of the corona radiata, corpus callosum and the superior longitudinal fasciculus. It was also noticed that patients with less severe course and those who stayed in hospital for a shorter time had less severe white matter abnormalities [201]. In a meta-analysis that included studies with a total group of 362 COVID-19 patients, WM abnormalities were observed in 20% of hospitalized group [202]. Damage to WM may have a hypoxic-ischemic cause, therefore the most vulnerable are patients with severe course of COVID-19 and requiring active oxygen therapy [203]. A PET CT study conducted among patients with COVID-19-related encephalopathy visualized hypometabolism of brain tissues in the PFC, ACC, insula (INS) and caudate nucleus (CN), which lasted up to 6 months after the disease [204]. Among COVID-19 patients, a reduction in the thickness of the GM of the orbitofrontal cortex (OFC) and the HPC together with an overall decrease in brain size was also found compared to healthy subjects [205].

Brain changes observed in BD include deviations in the size of the limbic structures, prefrontal, frontal and temporal lobes, WMH, connection and conduction disorders, loss of the GM thickness, and excessive activation of individual brain structures [185,186,187,188,196,197,198]. Some of the structural changes in the brain that occur during SARS-CoV-2 infection coincide with the changes observed in BD, e.g., anomalies of WM, thinning of the GM, and disorders in the PFC, ACC, INS and CN that persist many months after the disease [201,204,205]. It is also worth mentioning that the ACE2 receptor, which the virus can use to penetrate cells, is found on neurons of structures such as the thalamus, temporal lobe, CG and HPC [206]. Additionally, due to cytokines entering CNS and produced in it, neurotoxicity, neurodegeneration, inhibition of hippocampal neurogenesis and disorders of synaptic plasticity may occur [59,65,207,208]. These changes perhaps contribute to the degeneration of limbic structures and the signal transduction pathways between the PFC, FC and LS with subsequent hyperactivation of intact fibers, which may result in the characteristic symptoms of BD [199,200].

### 3.8. Neurotransmitters

Neurotransmission disorders are one of the early theories of affective disorders development. In the case of BD, anomalies in neurotransmission are also described. These include, among others glutamatergic and GABAergic disorders.

Glutamate (GLU) is the main excitatory neurotransmitter in the CNS. It acts via binding to various receptors, one of which is the NMDA receptor. At excitatory synapses, GLU released from neurons is taken up by astroglial cells and converted to glutamine (GLN) which then returns to neurons [209]. Moreover, GLU is a crucial factor in the production of glutathione (GSH), which is the most ubiquitous antioxidant in the brain [63]. GLU is also associated with nitrosative damage. Activation of NMDA receptors causes the activation of nitric oxide synthase (NOS) and the damage to DNA, lipids and proteins [63]. GLU is also a precursor of the principal inhibitory neurotransmitter—GABA [210]. The enzyme that converts glutamate to GABA is called glutamic acid decarboxylase (GAD) and it has two isoforms—GAD65 and GAD67 which are both present in the GABAergic neurons [211]. Magnetic resonance spectroscopy (MRS) revealed a significantly higher GLN/GLU ratio in the ACC and in the parieto-occipital cortex of patients with BD compared to the control group and schizophrenic patients which could be a sign of overactivity in the glutamatergic system or a disturbed glial nerve interaction [212]. A post-mortem study of patients with BD also detected increased GLU levels in their FC [213], and in a meta-analysis of studies using MRS, significantly increased levels of GLN and its derivatives were observed in structures such as the ACC, HPC, medial and dorsolateral PFC, parieto-occipital cortex and INS [214]. Hyperactivity of the glutamatergic system and too high concentration of GLU causes excitotoxicity, disturbances in neurotransmission and reduce cell viability [210,215]. The increase in GLU concentration can also be caused by the aforementioned QA [216]. Regarding the GABAergic system, inconclusive results are observed. Some researchers indicate reduced neurotransmission in the course of BD, as well as a reduced density of neurons containing GAD65 and GAD67 mRNA—especially in regions such as PFC, ACC, dentate gyrus and part of the HPC [217,218,219]. However, studies are inconclusive as some researchers report no differences in GABAergic neurotransmission between bipolar and control groups, and even indicate its increased activity and density of GAD65 and GAD67 mRNA neurons in the ACC and OFC regions [211,220,221]. GABA also plays a role in protecting cells against reactive oxygen species (ROS) by regulating the activity of various antioxidants. Thus, the reduced GABAergic activity in patients with BD described by some researchers could contribute to the damage in the CNS regions associated with the BD symptoms by reducing the antioxidant capacity [222,223].

Catecholaminergic disorders are also involved in the pathogenesis of BD. Especially dopaminergic and noradrenergic dysfunctions. These two systems share many common domains and overlap each other. Both systems fulfill similar physiological roles and are involved in wakefulness, arousal, memory formation and consolidation, and the reward system. NA binds to α1, α2 and β receptors, and DA binds to D1–D5 receptors. Additionally, the α2 receptor can be activated directly by DA, and dopaminergic and adrenergic receptors have the same stimulating pathways [224]. NA-producing nuclei are mainly located in the pons and medulla, and the greatest concentration of noradrenergic neurons is found in the locus coeruleus (LC). DA is formed mainly in the ventral tegmental area (VTA) as well as in the STR and LC [225,226]. It is believed that in patients with mania symptoms, the dopaminergic drive is over-activated, which subsequently downregulates its receptors and reduces sensitivity to DA—resulting in a depressive episode [227]. The confirmation of this theory may be the reduced availability of the DA receptors detected in the PET examination of people in the depressive phase of BD [228]. Patients receiving dopaminergic drugs, e.g., due to the Parkinson’s disease, may develop manic symptoms [229], and in patients with BD during a depressive episode, the administration of L-dopa could cause a manic shift [230]. Increased dopaminergic activity is also a source of oxidative stress in CNS. The products of DA metabolism by monoamine oxidase (MAO) are H2O2 and 3,4-dihydroxyphenylacetic acid, and the products of its non-enzymatic transformation are 6-hydroxydopamine (6-OHDA), which is a very reactive quinone. These compounds cause damage to proteins, DNA, lipids, amino acids and toxic damage to other parts of cells [231]. Simultaneously with the hyperactivation of the dopaminergic system, the closely related noradrenergic system is hyperactivated. In the manic phase, the concentration of NA and its metabolites in the urine and serum of patients with BD is higher than in the depressive phase and correlates with the severity of symptoms. Moreover, the concentration of NA in the urine increases even before the manic switch [232,233]. BD patients also have a higher number of catecholaminergic neurons in LC compared to MDD patients [234].

Serotonin is a neurotransmitter synthesized in the bodies of neurons and nerve terminals. There are 14 subtypes of serotonin receptors comprised in 7 main groups which are 5HT1–5HT7. Serotonergic fibers mainly come from two nuclei, the dorsal raphe and the median raphe, which send projections to the thalamus, hypothalamus, hippocampus, caudate, putamen and neocortex [235]. Thus, serotonin disorders affect many functions related to these regions, such as: sleep, sexual activity, appetite, impulse control, anger, aggression, memory and learning [236]. 5-HT dysfunction is widely reported in depression, however, it was noticed that case od mania and BD there is also a deficit in the level of 5-HT. One of the first hypotheses concerning 5-HT stated that in the case of depression and mania, serotonin levels decrease, and the clinical manifestation of mood disorders depends on hyper or hypoactivity of the dopaminergic and noradrenergic systems [237]. The evidence gathered so far seems to confirm 5-HT deficiency among BD patients at any stage of the disease [235,238]. However, not all researchers have found evidence of serotonergic hypoactivation in a manic state. Nikolaus et al. (2017) observed that, in patients with bipolar depression, there is an increase in 5-HT1R serotonin receptors in the HPC, pHPC and the AMG, as well as an increase in the serotonin transporter (SERT) in the CG and INS. On the other hand, in mania patients, 5-HT2R is reduced. These results may indicate serotonergic hyperactivity in the manic phase and hypoactivity in the depressive phase [239]. Studies in mice have also shown that serotonin deficiency causes aggressive and overly sexual behavior similar to that of patients in the manic phase [240].

Some evidence also points to cholinergic system involvement in the pathophysiology of BD. This system is involved in the control of the sensory processing, sleep, attention, arousal and higher cognitive functions [241]. Its neurotransmitter is acetylcholine which acts on two types of receptors: nicotinic (nAChR) and muscarinic (mAChR) [242]. Cholinergic neurons mainly originate in: brainstem, striatum, medial septum, nucleus basalis. Cholinergic projections are sent to the thalamus, olfactory bulb, neocortex, HPC and AMG. Moreover, cholinergic interneurons in the STR inhibit DA release [243]. It has been observed that patients in the depressive phase of BD have a reduced mean M2 mAChR binding in the ACC compared to patients with MDD and the control group. The binding appears to be correlated with the severity of depressive symptoms [244]. The lower number of M2 mAChR and their lower binding capacity, and according to some studies also the β2 nAChR, is probably the result of hyperactivation of the cholinergic system and the downregulation of its receptors during the depressive phase [245]. In contrast, the manic phase is connected to the hypoactivity of the cholinergic system. In many cases reported in the past, anticholinergic drugs have been shown to cause talkability and euphoria similar to those seen in mania [246]. In turn, drugs such as cholinesterase inhibitors and acetylcholine precursors in combination with lithium significantly reduce manic symptoms or depressive switch [247,248].

The possible effect of SARS-CoV-2 on neurotransmission may be related to, inter alia, the association of dopamine decarboxylase (DDC) with ACE2. DDC is the enzyme involved in the end-stage production of both DA and serotonin and shows the greatest co-expression and coregulatory relationship with ACE2. This relationship is not entirely clear, but it is possible that the downregulation of ACE2 by SARS-CoV2 may thereby influence the disorders of DA and serotonin synthesis [249]. Moreover, dopaminergic neurons show a high expression of ACE2, which may be directly related to a decrease in dopaminergic transmission [250]. Moreover, AT1 receptors, as well as D1 and D2, show counterregulatory expression, so downregulation of ACE2 by SARS-CoV-2 causing AT1 overexpression may result in decreased D1 and D2 expression and reduced DA sensitivity [251]. Possible changes in neurotransmission also apply to the glutamatergic system. In a case report of a patient with attention and memory disorders who underwent COVID-19, Yesilkaya et al. (2021) noticed, inter alia, a reduction in GLU levels and the GLU/GLN ratio in the patients’ dorsolateral PFC [252]. There is also evidence of the involvement of SARS-CoV-2 in cholinergic dysregulation [253]. SARS-CoV-2 spike protein (S protein) has a sequence similar to nAChR antagonists and can presumably bind to nicotinic receptors [254,255]. This is also important in relation to the cytokine storm in COVID-19 because one of the nicotinic receptors—α7 nAChR, which is found in substantial amounts, e.g., in the HPC, is the main receptor for the cholinergic anti-inflammatory pathway (CAP) and is involved in the modulation of pro-inflammatory cytokine production. The S protein binding to it will reduce its anti-inflammatory abilities [256,257,258]. There are also some reports of SARS-CoV-2 capabilities to cause GABA depletion which can results in decreased GABAergic inhibition in various parts of the CNS and decreased CNS anti-inflammatory resources [259,260,261].

To date, the results of research on neurotransmission disorders in COVID-19 indicate the possible influence of SARS-CoV-2 on neural signaling. However, the type of these changes, including decreased activation of the dopaminergic and glutamatergic systems corresponds more to those seen in MDD than in BD or mania [251,252]. The described serotonergic disorders are analogous to those described in BD and depression [249]. Cholinergic and GABAergic disorders seem to tip the side of those seen in BD, but more research is needed as the role of these two systems in the development of BD is still inconclusive. However, there is still little research on neurotransmission disorders in COVID-19, and to unequivocally assess its exact effect on neurotransmitters and the development of BD and other affective disorders, more detailed and in-depth analyzes are needed.

### 3.9. Oxidative Stress

Oxidative stress is an imbalance between oxidative and antioxidant factors in the body. Increased oxidative stress causing oxidation of proteins, nucleic acids and lipid peroxidation (LPO) has been well documented in the course of BD. The central nervous system (CNS) is particularly sensitive to this type of damage due to, among other things, the high content of substrates that can be oxidized, high oxygen tension and relatively low antioxidant capacity. These damages can lead to disturbances in neurotransmission, neuroplasticity and death of nerve cells [63,231,262]. The main enzymatic antioxidant factors are superoxide dismutase (SOD), glutathione peroxidase (GPx) and catalase (CAT). The major non-enzymatic factor is glutathione (GSH) [263,264]. Under normal circumstances, mitochondria are one of the main sources of ROS. The disturbed balance between antioxidant and oxidative factors causes damage to the mitochondria and a further increase in ROS production [231]. Other sources of increased oxidative stress in BD, as already mentioned, are: increased production of kynurenine metabolites, increased dopaminergic activity and decreased GABAergic activity. Oxidative stress and the cell death induced by it may be one of the causes of morphometric changes found, inter alia, in the frontal and subcortical regions of the brain of patients with BD [265]

In postmortem studies, higher concentrations of the markers of protein oxidation and nitration (carbonyls, 3-nitrotyrosine), and LPO (8-isoprostane, 4-hydroxy-2-nonenal, malondialdehyde) were found in the prefrontal cortex of BD patients [266]. Significantly increased level of LPO exponents, i.e., 4-hydroxynonenal (4-HNE) was also found in the ACC of BD patients [267]. A study using DTI and serum LPO index measurements indicted that WM abnormalities in BD patients are reflected in the serum by elevated lipid hydroperoxide (LPH) levels [268]. A meta-analysis of 27 studies on oxidative stress in BD found that the most frequent results were increases in nitric oxide level, LPO, and DNA/RNA damage [269]. One of the by-products of LPO are thiobarbituric acid reactive substances (TBARS), increased levels of which are detected in patients with BD, regardless of the phase of the disease (highest in the manic phase). In a study by Kapczinski et al. (2008) it has been shown that an increase in TBARS concentration correlates with a decrease in manic patients’ BDNF serum level which may be caused by the reduction in c-AMP response element-binding protein (CREB) by oxidative damage, energy deficiency and an increase in nuclear factor kappa-light-chain-enhancer of activated B cells (NF-κB) activity [270]. In addition, in the manic and depressive phase, the activity of SOD is also increased, and the activity of GPx is reduced. The SOD/GPx + CAT ratio is increased most in the manic phase and slightly less in the depressive phase [271]. Increased SOD, TBARS and CAT are a frequent findings, especially in untreated manic patients [272,273]. These results, however, are not unequivocal as there are also studies in which patients with BD in the acute phase of the disease, before starting treatment, had a reduced SOD and CAT activity, both in the depressive and manic phases. Moreover, SOD activity displayed an inverse correlation with the number of previous depressive episodes. This may indicate the exhaustion of antioxidant mechanisms in the course of the disease uncontrolled by proper medication [271,273,274,275,276]. In a postmortem study using the spectrophotometric method, it was found that the level of GSH in the prefrontal cortex of patients with BD and MDD is significantly reduced [277]. GSH and CAT reductions have also been demonstrated in living BD patients [278]. Other determinants of oxidative stress are DNA and RNA damages. Patients with BD, regardless of the stage of the disease, have significantly more DNA damage comparing to the group of healthy individuals [279]. RNA oxidative damage was also found in the HPC of BD patients in post-mortem studies [280]. A meta-analysis by Jiménez-Fernández et al. (2021) showed that patients with BD have significantly elevated levels of malondialdehyde (MDA), TBARS, CAT and glutathione transferase (GST), as well as significantly low level of GSH. Additionally, the level of TBARS was most strongly correlated with the manic state [37]

Neutrophils are cells that generate substantial amounts of ROS and reactive nitrogen species (RNS) which play an important role in fighting pathogens. However, too much activation of neutrophils results in the overproduction of ROS and RNS which cannot be compensated by the body’s antioxidant factors [281]. Among COVID-19 patients, a high neutrophil to lymphocyte ratio (NLR) and, consequently, excessive oxidative stress is observed [282]. It is also worth remembering that the ACE2 receptor is involved in inflammation and oxidative stress. Its substrate, angiotensin II, is a stimulant of NADPH oxidase (NOX) which is involved in ROS production. Moreover, ACE2 product—Ang (1–7) is a powerful antioxidant. Thus, downregulation of this receptor by SARS-CoV-2 could therefore increase oxidative stress by limiting antioxidant capacity and increasing ROS production [158,283]. These hypotheses are reflected in the results of the research which showed that COVID-19 (+) patients exhibit a NOX2 hyperactivity and intensified LPO and protein oxidation as compared to the control group. Furthermore, those factors were all positively correlated with the disease severity and mortality [284,285,286,287] Additionally, total antioxidant capacity proved to be lower and inversely correlated with disease severity [286,288]. In a small study by Lorente et al. (2021) a higher level of DNA and RNA oxidative damage was also observed in patients with COVID-19 who later passed away as compared to survivors [289]

In summary, patients with BD suffer from excessive oxidative stress and the damage it causes, such as LPO, oxidative damage to DNA, RNA and proteins, and to nitrosative damage [269,272,273]. Oxidative stress may damage brain structures involved in the pathogenesis of BD, such as the PFC, ACC, WM and HPC [266,267,268,280]. ROS-induced cell death may cause morphometric changes in the brain of BD patients [265]. COVID-19 can also generate oxidative stress, and its affinity for the ACE2 receptor diminishes the body’s antioxidant capacity and increases ROS production. In its case, intensified LPO, protein, DNA and RNA oxidation occur, and they are proportional to the severity of infection [158,286,287,289]. In the cytokine storm that takes place in COVID-19, elevated levels of cytokines such as TNF-α and IL-8, induce further ROS production. In turn, ROS causes the production of cytokines by activating the NF-κB pathway, thus creating a cycle that provokes serious cellular damage, probably also in the CNS, where it would contribute to the development of changes that predispose to the development of BD and other neuropsychiatric disorders [281,290,291]. However, to assess this, more research is needed to focus on the indicators of oxidative stress before and after developing COVID-19 and the possible subsequent development of affective disorders.

## 4. Discussion

In our study, we collected the previously reported cases of (hypo)mania that occurred in patients after infection with the SARS-CoV-2 virus, both in those who had never been treated for psychiatric diseases and in those with a psychiatric history [Table 1]. These descriptions are in many cases inconsistent, differ in the manner of examination and diagnostics, and access to all data is impossible. Therefore, it is difficult to talk about a reliable analysis of all the factors that could have caused a manic episode in the patients we collected. Nevertheless, we decided that the mere appearance of them shortly after infection with SARS-CoV-2 is a sufficient reason to analyze the similarities between the disorders caused by the infection and those that are present in patients with BD in order to assess whether SARS -CoV-2 may be a risk factor for developing this disorder. We consider this particularly important given the debilitating and progressive nature of BD and the still not fully understood effects of SARS-CoV-2.

We have divided the disorders of homeostasis that occur in patients with BD and its recognized risk factors into: (a) cytokine and inflammatory disorders, (b) use of corticosteroids, (c) HPA axis disorders, (d) sleep disorders, (e) BDNF deficiency, (f) hyperactivity of the kynurenine pathway, (g) structural changes in the CNS, (h) neurotransmission disorders, (i) oxidative stress. These factors create a complex system in which most of them interact with each other, creating a self-reinforcing cycle that results, as we argue, in the development of affective disorders—including BD. Our observation shows that SARS-CoV-2 infection affects most of the above-mentioned factors with varying severity. In our opinion, the most visible disorder that occurs in SARS-CoV-2 and that may affect the development of BD in patients is systemic inflammation and cytokine storm. In the case of SARS-CoV-2, an increase in the concentration of inflammatory cytokines such as IL-1β, IL-2, IL-6, IL-7 and TNF-α comes to the fore, along with the rise in IL-4 and IL-10 concentrations in a later stage of infection [70,72,74,75]. Thus, Th1 activation occurs, followed by Th2 activation. A similar cytokine profile is found among BD patients. They also have an increase in the concentration of inflammatory cytokines, and some researchers also note the activation of Th2 and an increase in the concentration of anti-inflammatory IL-4, IL-5 and IL-10 in patients in the manic phase [63,65,67,81]. Although the exact meaning and mechanism of the influence of anti-inflammatory cytokines on the development of manic symptoms has not been studied, this similarity between SARS-CoV-2 and BD may be significant. As far as inflammatory cytokines are concerned, they cause a decrease in synaptic activity, hippocampal neurogenesis, degeneration of limbic structures and disturbances in neural signaling, in addition, they influence the activation of the kynurenine pathway and the HPA axis, ROS production, CNS structural changes and indirectly affect neurotransmission [51,52,57,58,59]. Another clearly visible disorder in COVID-19 is the activation of the kynurenine pathway through increased activation of IDO under cytokine storm conditions. In addition, the cytokine profile in COVID-19 patients contributes to a stronger activation of the microglial IDO variant, the end products of which are mainly 3-HK and QA [177,179,181]. These are metabolites that cause neurodegeneration, excitotoxicity and ROS generation. BD also increases the production of kynurenine and its metabolites, and in the case of mania it is precisely the high concentration of 3-HK and QA that occur in patients [172,173]. The properties of kynurenine metabolites may contribute to cell death and cell signaling disturbances in brain regions associated with BD in COVID-19 patients. Another disorder that has a strong similarity between SARS-CoV-2 and BD patients is oxidative stress. In both cases, there is an increase in the level of LPO and oxidative and nitrosative damage to proteins and RNA, as well as the depletion of antioxidant systems [269,272,273,287,289]. In patients with BD, this can occur over an extended period of time, while in patients with SARS-CoV-2 the process is quite rapid. The very use of the ACE2 receptor by the virus causes a reduction in systemic antioxidant resources and an increase in ROS production, which increases even after the onset of a cytokine storm [158,283]. This causes damage to the nerve structures, signal transduction pathways and morphometric changes in the brain of patients that can initiate symptoms of BD, especially when these changes occur rapidly as in the case of SARS-CoV-2. Brain disorders in patients with SARS-CoV-2 similar to those in patients with BD and which may influence the development of its symptoms are mainly WM abnormalities. They may affect signaling pathways between the PFC, FC and LS and cause a subsequent hyperactivation of intact fibers which may result in the characteristic symptoms of BD [201,204,205]. The above-mentioned inflammatory cytokines, kynurenine metabolites and oxidative stress contribute to the development of WM anomalies and disorders of nerve signal transduction. Another common point of BD and SARS-CoV-2 is sleep disturbance. Not only is insomnia one of the symptoms of mania, but it can also contribute to its development. Sleep disorders, on the other hand, are one of the more common complications of COVID-19 reported by patients. Their influence on the development of BD symptoms may include the activation of the HPA axis and the induction of the secretion of neurotransmitters such as DA, GABA and GLU [3,135,136,137,138]. Disturbances in neurotransmission are widely described in BD. In mania, they mainly concern catecholaminergic and glutamatergic hyperactivity as well as GABAergic, cholinergic and serotonergic hypoactivity [212,232,233,246,249]. Changes in neurotransmission in COVID-19 are not well understood and remain hypothetical. There are hypotheses regarding SARS-CoV-2 capabilities to lower dopaminergic, cholinergic, GABAergic and serotonergic transmission, which would result in a picture resembling that of depression, not mania [251,252]. However, when considering the risk of BD, it should be remembered that the depressive phase also belongs to the disease picture. Another factor with no definite answer is BDNF. In BD there is an overall decrease in its level, although there is a theory that the mania state itself is caused by temporary BDNF overactivity, which may be confirmed by the formation of manic symptoms after sleep deprivation followed by a sudden increase in BDNF [143,144,145,146,150]. In COVID-19 there is mainly a decrease in BDNF level, and this may be due to the affinity of SARS-CoV-2 for the ACE2 receptor, which is involved in the BDNF formation pathway [155,156,157,158]. However, in the case of patients who developed “long-COVID”, which also included cognitive disorders, there is a visible increase in its level [160]. We therefore believe that the rapid clinical improvement of patients and the subsequent rapid normalization of BDNF or the temporary overactivity of BDNF resulting from sleep disorders could mediate the development of mania. In the case of the HPA axis, it is also difficult to come to unambiguous conclusions. In BD, especially in the manic phase, hyperactivation of the HPA axis is more visible, especially in people with a short course of the disease [105]. On the other hand, studies in COVID-19 are ambiguous and, in addition to descriptions of hypercortisolemia, there are also reports of hypocortisolemia and AI [110,111,114,115,116]. It cannot be ruled out that in patients who develop an increase in cortisol, it may be associated with an increased risk of mania, especially when there are large diurnal fluctuations in cortisol. The last factor, which is the only external one and is not a disorder caused by SARS-CoV-2, is the effect of treatment with corticosteroids. Their mania-inducing effect is generally recognized, and the doses and cycle duration recommended for the treatment of COVID-19 coincide with those that pose a high risk of developing manic symptoms [85,86,91,92,98]. In the cases we collected, 38.8% of patients received steroids and we cannot exclude that steroids were the cause of mania in these patients, but in more than 60% they were not included in treatment, which may indicate a different mechanism of mania development [Table 1].

As can be seen, many of the disorders caused by SARS-CoV-2 overlap with those found in BD patients [Figure 1]. However, their ability to induce mania or full-blown BD in previously healthy and unburdened individuals may not be as significant. Looking at the statistics, we can see that affective disorders concern from 4 to almost 14%, and newly diagnosed cases account for about 4% of patients [3]. So far, there are no data limited to cases of mania, but assuming that depressive episodes occur much more often, it can be concluded that mania is only a small fraction of the previously mentioned 14 and 4% of patients. Comparing this with the lifetime risk of BD, assessed from 0.1% to 2.4%, we can see that COVID-19 related mania or BD cases do not go beyond this risk [11]. However, most cases of BD begin with depressive episodes, which are much more common than mania after COVID-19, and a definitive diagnosis of BD may not be made for many years after the first affective episode. In addition, depressed patients are more susceptible to other risk factors for the development of BD, and SARS-CoV-2 may sensitize patients to those factors and thus contribute to the development of BD. We also believe that the disorders caused by SARS-CoV-2 are so similar to those seen in BD that, even if they are insufficient to trigger an episode in previously healthy individuals, they may be a serious risk factor for relapse in people currently in euthymia or accelerate the onset of the disease in people who are particularly vulnerable to it, e.g., genetically burdened or who have a family history of this disorder. It is also worth noting that the cases of mania that have been associated with COVID-19 in the literature concern patients whose symptoms appeared up to 3 weeks after the diagnosis of the infection [Table 1]. Additionally, some disorders caused by COVID-19 may last up to several months, and although the appearance of affective symptoms will no longer be associated with SARS-CoV-2, prolonged disorders may contribute to their appearance [201,204].

Clinicians should also pay attention to patients with more severe COVID-19 because most of the changes we have described are proportional to the severity of the coronavirus disease, and more severe lesions increase the risk of developing BD. In addition, corticosteroid therapy, a recognized risk factor for mania, should be limited to those really requiring it. In patients with the highest risk of developing an affective episode, e.g., seriously ill patients with BD, currently in euthymia, a control of individual disorders caused by SARS-CoV-2 may be considered, e.g., taking care of patients’ sleep hygiene—melatonin supplementation may prove to be helpful and is sometimes used as a adjunctive treatment in BD [292,293], support of antioxidant systems—in some cases, supplementation with N-acetylcysteine proves to be useful, especially in relation to depressive symptoms BD [294,295]. In patients with severe cytokine storm, the use of tocilizumab may be helpful [296,297], although its effect on the symptoms of BD is inconclusive [298]. In the future, in order to better assess the influence of SARS-CoV-2 on the development of BD, there is a need for prospective studies evaluating the effect of the virus on the systems and factors contributing to the pathogenesis and course of BD, as well as longer observation of patients for the emergence of affective disorders, in particular episodes of mania or hypomania. Additionally, future research may attempt to consider the association of SARS-CoV-2 with various forms of BD such as rapid cycling, childhood and adolescent BD or unipolar mania, which we did not include in our work due to the current lack of relevant literature.

## 5. Conclusions

In the course of SARS-CoV-2 infection, patients develop cytokine disorders, activation of the kynurenine pathway, increased oxidative stress, structural changes in the CNS and sleep disorders. All of these factors largely overlap with the disorders and changes seen in people with BD. In the course of infection, there are also disturbances in neurotransmission and regulation of the HPA axis, as well as a decrease in the level of BDNF—these changes, however, are less consistent with those in BD patients. Due to all the abovementioned factors, infection with SARS-CoV-2 and COVID-19, especially with severe course, may be a risk factor for the development of BD in particularly vulnerable people, e.g., with a family history, or lead to relapse in previously diagnosed BD patients. An additional risk factor is corticosteroid therapy during hospitalization of infected patients. Clinicians should be aware of the possibility of developing affective episodes, including mania, in vulnerable patients and extend the follow-up period after discharge from hospital as some post-COVID depression may turn out to be misdiagnosed BD over time.

## Figures and Tables

**Figure 1 jcm-11-06060-f001:**
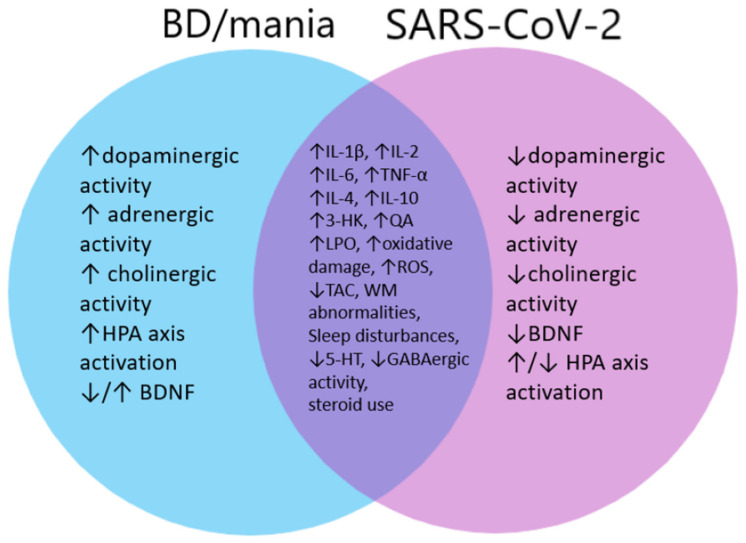
Convergence of disorders caused by SARS-CoV-2 and factors contributing to the development and course of BD/mania. Disorders inside the common part of both circles coincide with each other the most. Disorders outside the common part of the circles are different, inconclusive or inconsistent (explained in the main text). Abbreviations: ↑—increased concentration/activity; ↓—decreased concentration/activity; 3-HK—3-hydroxykynurenine; 5-HT—serotonin; BD—bipolar disorder; BDNF—brain-derived neurotrophic factor; GABA—γ-Aminobutyric acid; HPA—hypothalamic-pituitary-adrenal; IL-1β—interleukin-1β; IL-2—interleukin-2; IL-4—interleukin-4; IL-6—interleukin-6; IL-10—interleukin-10; LPO—lipid peroxidation; QA—quinolinic acid; ROS—reactive oxygen species; TAC—total antioxidant capacity; TNF-α—tumor necrosis factor α; WM—white matter.

**Table 1 jcm-11-06060-t001:** Cases of (hypo)mania related to COVID-19. Abbreviations: ↑—increase; ↓—decrease; (−)—negative; (+)—positive; (l.d.)—low dose; (u.d.)—unknown dose; N—normal/no change; -—results not available; M—male; F—female; BD—bipolar disorder; CRP—C reactive protein; CSF—cerebrospinal fluid; ESR—erythrocyte sedimentation rate; IL-6—interleukin-6; IL-10—interleukin-10; inhaled steroids; K^+^—potassium ions; LDH—lactate dehydrogenase; PTSD—post traumatic stress disorder; SLE—systemic lupus erythematosus; WMH—white matter hyperintensities.

Authors	Patient	Diagnosis	SARS-CoV-2 Confirmed ^1^	COVID-19 Severity ^2^	Time ^3^	Symptoms	Use of Steroids	Laboratory Findings	CSF Findings	Changes in Neuroimaging	Psychiatric History
Lu et al. [25]	51 M	Acute manic episode	Yes	Mild	17 days	Insomnia, agitation, irritability, grandiosity, increased energy, talkativeness	Yes (u.d)	Leukopenia ↑IL-6 ↑IL-10 ↑CRP	SARS-CoV-2 RNA (−), specific IgG antibody (+)	N	No
Sen et al. [27]	33 F	Acute manic episode	No (clinical picture)	Asymptomatic	-	Insomnia, agitation, irritability, delusions, talkativeness	No	Leukocytosis ↑CRP ↑fibrinogen ↑ferritin ↑D-dimer	-	WMH in the corpus callosum, possible cytotoxic edema	No
Correa-Palacio et al. [41]	43 M	Steroid-induced manic episode	-	Moderate	12+ days	Insomnia, agitation, irritability, aggression, grandiosity, delusions, hallucinations	40 mg ^4^	-	-	N	No
Park et al. [24]	56 M	Acute manic episode	Yes	Mild	7 days	Insomnia, agitation, sexual disinhibition, delusions, distractibility, increased energy	No	↑ESR ↑CRP neutrophilia thrombocytosis	-	N	No
Mawhinney et al. [26]	41 M	Acute manic episode	Yes	Moderate	10 days	Euphoria, agitation, sexual disinhibition, grandiosity, talkativeness	No	-	N	N	No (BD in family)
Haddad et al. [45]	30+ F	Delirious mania	Yes	Mild	0–2 days	Insomnia, euphoria, agitation, sexual disinhibition, grandiosity, hallucinations, distractibility, talkativeness	No	↑CRP ↑ferritin ↑LDH ↑IL-6	-	-	No
Varsak et al. [31]	64 F	Acute manic episode	No (clinical picture)	Moderate	12 days	Euphoria, irritability, aggression, delusions, hallucinations, talkativeness	Yes (inh.)	-	-	N	No
Shanmugam et al. [33]	52 M	Acute manic episode	Yes	Moderate	14 days	Euphoria, agitation, sexual disinhibition, grandiosity, talkativeness	No	N	-	N	No
Iqbal et al. [30]	Mean: 40 14 M, 1 F	12/15 Acute manic episode, 3/15 Acute hypomanic episode	Yes	10 Asymptomatic 2 Mild 1 Moderate 2 Severe	-	13/15 Insomnia 10/15 euphoria 9/15 delusions 9/15 irritability 8/15 agitation 8/15 aggression 5/15 distractibility 5/15 hallucinations	3/15 (u.d.)	7/15 ↑CRP (n = 4) Neutrophilia (n = 3)	-	3/15 mild white matter ischemic changes	Yes 8/15 (6/15 previous mania)
Noone et al. [34]	34 F	Acute manic episode	Yes	Asymptomatic	17 days	Insomnia, agitation, irritability, delusions, distractibility, talkativeness	No	N	N	Subcortical WHM	No
Kurczewska et al. [47]	44 M	Bipolar disorder	Yes	Severe	15–17 days	Insomnia, euphoria, agitation, sexual disinhibition, grandiosity, delusions, distractibility	Yes (u.d.)	↑CRP ↑ESR	-	Arachnoid cysts	No
Uzun et al. [35]	16 M	Acute manic episode	Yes	Mild	+10 days	Insomnia, euphoria, irritability, increased energy, talkativeness	-	-	-	-	No
Kashaninasab et al. [32]	25 M	Acute manic episode	No (clinical picture)	Mild	-	Insomnia, agitation, irritability, aggression, sexual disinhibition, grandiosity, delusions, talkativeness	-	-	-	-	-
Jasani et al. [42]	36 F	Steroid-induced mania	Yes	Mild	-	Insomnia, euphoria, distractibility, improper reactions	Yes 30–40 mg	Leukopenia ↑CRP ↑ESR	Low protein, lymphocytosis	N	No (SLE)
Kazi et al. [43]	55 F	Steroid-induced mania	Yes	Moderate	21 days	Insomnia, agitation, talkativeness, grandiosity, delusions, improper judgment and behavior,	Yes 40 mg	↑K^+^ ↑LDH ↑CRP ↑ferritin	-	N	No
Perez et al. [44]	39 M	Steroid-induced mania	Yes	Moderate	23 days	Insomnia, grandiosity, religious delusions euphoria, talkativeness, irritability	Yes 40 mg	N	-	-	Depression, PTSD, anxiety
Grover et al. [36]	25 M	Mania	Yes	Moderate	14 days	Insomnia, euphoria, grandiosity, irritability, talkativeness	Yes 40 mg	N	-	N	No (Depression in family)
Jimenez-Fernandez [48]	71 M	Mania	Yes	Mild	0 days	Insomnia, agitation, euphoria, grandiosity, sexual disinhibition, talkativeness, cognitive disorders	Yes (l.d.—not for COVID)	Leukocytosis Neutrophilia ↑CRP	-	Cortico-subcortical atrophy	No
Russo et al. [38]	60 F	Acute manic episode	Yes	Asymptomatic	2 days	Insomnia, agitation, euphoria, aggression, delusions, hallucinations, talkativeness, anxiety	Yes 1 mg/kg (2 days)	N	N	N	Depression, (Depression, delusions in family)
Uvais et al. [39]	22 F	Acute manic episode	Yes	Asymptomatic	2 days	Insomnia, agitation, grandiosity, irritability, sexual disinhibition, talkativeness	No	-	-	-	Depression, OCD
Uvais et al. [46]	36 M	Delirious mania	Yes	Mild	7 days	Insomnia, agitation, grandiosity, irritability, talkativeness, confusion, loss of appetite	No	N	N	N	No
Uvais [40]	45 F	Acute manic episode	Yes	Severe	21 days	Insomnia, agitation, grandiosity, irritability, talkativeness	Yes (u.d.)	↑D-dimer	-	-	Depression (BD in family)

^1^ SARS-CoV-2 confirmed by a RT-PCR test or positive antibody test, if not—clinical picture indicated COVID-19. ^2^ Severity of COVID-19 was assessed according to WHO guidelines. ^3^ Interval between COVID-19 first symptoms and onset of manic symptoms. ^4^ All doses of corticosteroids were converted to corresponding doses of prednisone.

## Data Availability

No new data were created or analyzed in this study. Data sharing is not applicable to this article.

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
