# Peer review of "Is SARS-CoV-2 a Risk Factor of Bipolar Disorder?—A Narrative Review"

_jcm, 2022, doi:10.3390/jcm11206060_

Round 1

Reviewer 1 Report

The present review aims to address the knowledge gap in understanding the neurobiological links between COVID and bipolar disorder. The inclusion of various neurobiological factors with the latest evidence can be considered one of the strengths of the paper. Appropriate references are included in the review. 

Revising it to have a specific focus on COVID alone has enhanced the clarity of the manuscript. 

Some of the minor comments are as follows:

-Kindly insert the year in brackets adjacent to Parboosing et al. and other similar references

- Have the authors attempted to look at the association or link between COVID and various forms of BD (rapid cycling, unipolar mania, childhood or adolescent BD, etc.) OR they have restricted their observations to general descriptions of BD seen in adults? If such variants are not analyzed, they can either include them as limitations or consider them as avenues for future research. 

Author Response

The authors would like to thank the Reviewer 1 for valuable comments towards improving our manuscript. We deeply appreciate the Reviewer 1 assistance. All comments and suggestions of the Reviewer have been carefully read and taken into consideration during preparation of the revised version. All changes mad in the manuscript according to the Reviewers' opinions have been written in red.

Below we answer the comments raised by the Rewiever 1:

-Kindly insert the year in brackets adjacent to Parboosing et al. and other similar references - we have made the corrections to all references according to the Rewiever's comment

  • Have the authors attempted to look at the association or link between COVID and various forms of BD (rapid cycling, unipolar mania, childhood or adolescent BD, etc.) OR they have restricted their observations to general descriptions of BD seen in adults? If such variants are not analyzed, they can either include them as limitations or consider them as avenues for future research. - According to the Reviewer's suggestion, we have addressed that issue in the 'Discussion' section, lines 1066-1069.

Once more, we thank the Reviewer 1 for the review of our manuscript and all of the helpful comments.

Reviewer 2 Report

The authors focused on a very important topic in psychiatry, such as
whether SARS-CoV-2 is a risk factor for bipolar disorder?.
In fact, 299 references were used in the preparation of the manuscript,
which may indicate the great diligence of the authors in the in-depth study
of the topic they decided to choose.
The reviewer noticed some shortcomings of the manuscript, despite the fact
that the text was written with the preservation of scientific manner and in respect
to the
journal guidelines.
Therefore, the authors are asked to take into consideration the comments of
the reviewer listed below and respond to them:
  • Missing information about how many publications were taken into direct analysis.

  • Line 115-120 (quote) “ (…) we have identified the most frequently described changes and biomarkers of BD, as well as disorders detected in SARS- CoV-2 infection and divided them into: inflammatory factors, the effect of medications, disorders of the hypo-thalamic-pituitary-adrenal axis (HPA) , sleep disorders, growth factor disorders, kynurenine pathway activation, structural changes in the central nervous system (CNS), disturbances in neurotransmission and oxidative stress. “ It seems cases of (hypo)mania is omitted as part of raised findings in the above quoted and next described in domain “3.1. Cases of (hypo)mania “.

  • Line 124-125 (quote) “ 12 patients were previously diagnosed with psychiatric illness, 6 of which experienced manic episode in the past “. In the quoted phrase missing numerical resources of the raised information, and to support such information.

  • Line 128-129 (quote) “ In 3 patients, SARS-CoV-2 infection was not confirmed by a formal test, but only on the basis of the clinical picture and characteristic imaging changes in the lungs.” In the quoted phrase missing numerical resources of the raised information, and to support such information.

  • Not all abbreviations used in the text for the first time were explained
    by the authors.

Author Response

The authors would like to thank the Reviewer 2 for valuable comments towards improving our manuscript. We deeply appreciate the Reviewer 2 assistance. All comments and suggestions of the Reviewer have been carefully read and taken into consideration during preparation of the revised version. All changes made in the manuscript according to the Reviewers' opinions have been written in red.

Below we answer the comments raised by the Rewiever 2:

  • Missing information about how many publications were taken into direct analysis - According to the Reviewer comment, we have provided that information in 'Results' section, lines 113-115 and 144-146.
  •  It seems cases of (hypo)mania is omitted as part of raised findings in the above quoted and next described in domain “3.1. Cases of (hypo)mania - We have incorporated a change in layout of the manuscript so that the description of hypo(manic) cases would be a part of main results as well as we indicated that in the text, line 143.
  • Line 124-125 (quote) “ 12 patients were previously diagnosed with psychiatric illness, 6 of which experienced manic episode in the past “. In the quoted phrase missing numerical resources of the raised information, and to support such information - corrected, line 118

  • Line 128-129 (quote) “ In 3 patients, SARS-CoV-2 infection was not confirmed by a formal test, but only on the basis of the clinical picture and characteristic imaging changes in the lungs.” In the quoted phrase missing numerical resources of the raised information, and to support such information - corrected, line 123

  • Not all abbreviations used in the text for the first time were explained
    by the authors 
    - According to the Reviewer's comment, we have explained every missing abbreviations

Once more, we thank the Reviewer 2 for the review of our manuscript and useful comments.